# A global semi-empirical GIA model based on GRACE data

Yu Sun[1], Riccardo E. M. Riva[2]

[1]Key Laboratory of Data Mining and Sharing of Ministration of Education, Fuzhou University, Fuzhou, China.
[2]Dept. of Geoscience and Remote Sensing, Delft University of Technology, Delft, The Netherlands

*Correspondence to*: Riccardo E. M. Riva (r.e.m.riva@tudelft.nl)

**Abstract.** The effect of Glacial Isostatic Adjustment (GIA) on the shape and gravity of the Earth is usually described by numerical models that solve for both glacial evolution and Earth's rheology, being mainly constrained by the geological evidence of local ice extent and globally distributed sea level data, as well as by geodetic observations of Earth's rotation.

In recent years, GPS and GRACE observations have often been used to improve those models, especially in the context of
regional studies. However, consistency issues between different regional models limit their ability to answer questions from global scale geodesy. Examples are the closure of the sea level budget, the explanation of observed changes in Earth's rotation, and the determination of the origin of the Earth's reference frame.

Here, we present a global empirical model of present-day GIA, solely based on GRACE data and on geoid fingerprints of mass redistribution. We will show how the use of observations from a single space-borne platform, together with GIA fingerprints
based on different viscosity profiles, allows us to tackle the questions from global scale geodesy mentioned above. We find that, in the GRACE era (2003-2016), freshwater exchange between land and oceans has caused global mean sea level to rise by $1.2 \pm 0.2$ mm/yr, the geocentre to move by $0.4 \pm 0.1$ mm/yr, and the Earth's dynamic oblateness ($J_2$) to increase by $6.0 \pm 0.4 \ 10^{-11}$/yr.

## 1    Introduction

The observation-based estimation of mass redistribution in the Earth's water layer from regional to global scales has been made possible in the last two decades by the Gravity Recovery and Climate Experiment (GRACE) satellite mission (Tapley et al., 2004; Wouters et al., 2014).

However, since observations of time-variable gravity are intrinsically sensitive to any mass change, the contribution of the solid earth needs to be removed. In particular, it is necessary to account for the effect of a few great earthquakes (Han et al.,
2013) and of glacial isostatic adjustment (GIA). The latter represents the delayed viscoelastic response of the Earth to past glacial cycles (Peltier, 2004), and it is the only process relevant at global scales.

Historically, GIA has been investigated by means of numerical models that solve for both changes in the ice cover over a glacial cycle and for the Earth's mechanical properties, in particular mantle viscosity. Those models are mainly constrained by the geological evidence of past ice extent, by reconstructions of past sea level change, and by observations of earth rotation
(Peltier, 1982; Nakada and Lambeck, 1987; Mitrovica et al., 2015; Nakada et al., 2017). While those models aim at

understanding past glaciations and their effect on sea level and earth rotation, they might not be optimal for providing an accurate correction for the solid earth contribution to GRACE observations. This is mainly due to the fact that available observations are sparse in both space and time, which largely limits the complexity of GIA models, hence their accuracy. In order to improve the ability of GIA models to reproduce present-day signals, they have been further constrained by geodetic

observations of vertical land motion (Peltier et al., 2015; Caron et al., 2017). Nonetheless, GIA model uncertainties are still one of the main source of errors for, e.g., GRACE-based estimates of global mean ocean mass change (WCRP Global Sea Level Budget Group, 2018).

An alternative approach to model the effect of present-day GIA makes use of satellite-based geodetic observations in order to generate empirical (or data-driven) models. So far, those models have been tailored to regions that were covered by the largest

ice sheets, namely Antarctica, Northern Europe and North America (e.g., Riva et al., 2009; Hill et al., 2010; Simon et al., 2017). Regional models allow to obtain an improved accuracy by relying on multiple datasets (e.g., GPS, GRACE, satellite altimetry), without introducing consistency issues that usually arise when working with satellite data at global scales, such as the problem of assuring mass conservation or of using a common reference frame. As a result, those models typically do not allow to properly tackle global problems, such as the determination of total ocean mass change.

Here, we present results from a semi-empirical GIA model solely based on GRACE data and on physical basis functions, represented by geoid fingerprints of known sources of mass change. The fingerprint approach used in this study has been initially proposed by Rietbroek et al. (2012) for sea level studies, and adapted by Sun et al. (2019) to study changes in the Earth's oblateness, where one of the main differences is that the approach by Rietbroek et al. (2012) made also use of data from satellite altimetry. This is the first time that the current approach is used to specifically produce GIA model results. We

consider our result to be a semi-empirical model because it makes use of GIA fingerprints that are based on the same physics as the classical forward models, but that are afterwards tuned to match present-day observations of gravity changes, without providing updated estimates of past ice change and/or mantle viscosity.

## 2 Methods

### 2.1 Inversion

The method has been discussed in Sun et al. (2019). In summary, we construct 151 fingerprints of geoid change induced by unit mass variations of continental water, solving the sea level equation (Farrell and Clark, 1976) on a compressible elastic earth based on the Preliminary Reference Earth Model (PREM, Dziewonski and Andersen, 1981), and including the rotational feedback (Milne and Mitrovica, 1998). The fingerprints are based on: individual drainage basins for the two ice sheets, glacier regions from the Global Land Ice Measurements from Space (GLIMS) database (Kargel et al., 2014) and empirical orthogonal

functions of land hydrology (Rietbroek et al., 2016). Those fingerprints are fundamentally the same as in Sun et al. (2019), with minor updates: over the ice sheets, we have merged a few neighbouring drainage basins that were providing anti-correlated solutions (Antarctica: next to the East Antarctic Weddell Sea, and on the Northern Peninsula; Greenland: in the North East

interior), we do not model peripheral glaciers around the Greenland Ice Sheet, and we have added separate fingerprints for the Southern Patagonia Ice Field and for Lake Victoria.

In addition, we define 132 sets of seven fingerprints of geoid change induced by GIA: six over distinct sub-regions (as in Sun et al., 2019: three for North America, two for Northern Europe, one for Antarctica), and an additional fingerprint for the effect of GIA-induced changes in the position of the Earth's rotation axis (True Polar Wander). More detail about the GIA fingerprints is given below. Note that, in Sun et al. (2019), we made use of a single set of six regional GIA fingerprints.

Through a least-square approach in the spectral domain, we simultaneously fit all fingerprints to CSR RL06 GRACE monthly
fields of geoid height changes, expanded up to spherical harmonic degree 60 and ranging from January 2003 to August 2016 (Save et al., 2018), with the additional constraint that the GIA fingerprints have to follow a linear trend (i.e., that the GIA monthly variations are constant through the whole timespan). The result is a time series of scaling factors that, once multiplied by the respective fingerprints and added together, optimally reproduces the original GRACE fields. It is important to notice that we only use GRACE spherical harmonic coefficients starting from degree 2 and order 1: in other words, we do not force
the solution to fit GRACE observations of changes in the Earth's oblateness, as it will be discussed later.

The obtained set of scaled fingerprints provides sufficient information for partitioning the total GRACE signal into a number of components, driven by independent processes: GIA, which is the main object of this study, as well as mass changes in the cryosphere and in land hydrology. The ocean is considered to be passive, meaning that we assume the effect of internal mass redistribution by ocean dynamics to be accurately removed by the ocean de-aliasing products used during GRACE data
processing (Dobslaw et al., 2017).

We perform the inversion independently for each of the 132 sets of GIA fingerprints, hence generating an ensemble of 132 solutions of GIA and of the effect of continental water mass redistribution. The ensemble mean and its standard deviation represent the final solution.

## 2.2 GIA fingerprints

GIA fingerprints are obtained from solving the sea level equation for a spherically symmetric, viscoelastic, incompressible and non-rotating PREM earth (Kendall et al., 2005; Martinec et al., 2018). We have chosen to use an incompressible earth model, because the induced gravity changes are very similar to those of a compressible earth (Tanaka et al., 2011), but it is computationally more stable.

A forward GIA model requires to define an ice history and an earth model. As ice history, we use either GLAC1D (Tarasov et
al., 2012) and ANU (Lambeck et al., 2010) in North America and Northern Europe, respectively, or ICE-6G_C (Peltier et al., 2015) in both regions. Concerning the earth models, we use a 100-km-thick elastic lithosphere together with all possible combinations of 6 viscosity values in the upper mantle (range 1e20-1e21 Pa s) and 11 viscosity values in the lower mantle (range 1e21-1e23 Pa s), giving rise to 66 variants. Together with the two alternative ice histories for the Northern Hemisphere, we obtain 132 sets of GIA fingerprints.

For Antarctica, we use a single fingerprint, based on ice history IJ05 (Ivins and James, 2005), in combination with a 65-km-thick elastic lithosphere, and a viscosity of $5 \times 10^{20}$ Pa s and $10^{22}$ Pa s in the upper and lower mantle, respectively. This Antarctic set-up showed very good agreement with the empirical GIA model of Riva et al. (2009).

For Greenland, we have no dedicated fingerprint due expected small signals and to their spatial overlap with the signature of present-day ice mass changes, hence we only account for the far-field effects of the former neighbouring ice sheets.

Finally, we produce a GIA-induced polar motion fingerprint by first performing a preliminary inversion where we use the six regional GIA fingerprints, generated without rotational feedback. We then take the six resulting pairs of degree 2 order 1 coefficients and add them together to form a new GIA-induced polar motion fingerprint. In the final inversion, we set the degree 2 order 1 coefficients of the six regional GIA fingerprints to zero, and we treat the GIA-induced polar motion fingerprint separately (albeit with a fixed C21/S21 ratio, as determined in the preliminary inversion).

## 2.3   Low-degree solutions

As discussed in Sun et al. (2019) and mentioned above, the least-squares solution makes only use of GRACE observations from spherical harmonic degree 2 and order 1. However, the GIA fingerprints are complete from degree 2 order 0, while the land water fingerprints are complete from degree 1 order 0. Hence, even if observational constraints are not directly applied, the fact that a single scaling factor is determined for each fingerprint implies that the inversion can also provide a solution for

the Earth's oblateness ($J_2$, related to degree 2 order 0) and geocentre motion (degree 1). $J_2$ estimates are important because its observations from GRACE are notoriously poor (Chen and Wilson, 2008), while geocentre motion cannot be directly observed by GRACE, but it is necessary to accurately determine mass changes in the Earth's water layer (Chen et al., 2005).

Finally, secular polar motion is particularly interesting, since there is still no consensus in the community about the exact implementation of the rotational feedback in GIA modelling (Peltier and Luthcke, 2009; Mitrovica and Wahr, 2011; Martinec

and Hagedoorn, 2014). Considering that the impact of mass redistribution in the water layer on polar motion is an integral part of the elastic fingerprints, the use of a separate fingerprint for the effect of GIA-induced polar motion means that we let it be scaled by GRACE degree 2 order 1 observations, under the assumption that the rotational feedback will affect polar motion magnitude and direction rather than orientation (visually confirmed from Milne and Mitrovica, 1998).

## 3   Results

In Fig. 1, we show the ensemble GIA solution and its standard deviation, both represented in terms of geoid height changes, consistently with the GRACE input.

In North America, the largest values are obtained over Hudson Bay, whereas the largest uncertainties can be found in the neighbouring regions, in particular SE of Lake Winnipeg and SE of Baffin Island. Over Northern Europe, the largest values as well as the largest uncertainties are found over the Gulf of Bothnia. Notably, the ensemble solution does not show any

significant signal over the Barents Sea, apart from a NE extension of the 0.1 mm/y contour to include Novaya Zemlya, reflecting the absence of large signals in the input GRACE fields.

The solution over Antarctica, and in particular its minimal uncertainty, is a direct result of the use of a single fingerprint: in principle, the approach allows for a variable Antarctic scaling, depending on the impact of alternative GIA fingerprints in the northern hemisphere, but in practice the solution is dominated by the near-field regions.

Finally, a clear signal originates from secular polar motion, which causes a positive trend over Central Asia and southern South America, a negative trend over the southern Indian Ocean, and a southern extension of the peripheral bulge over Central America. Quantitatively, our solution for GIA-induced polar motion points towards $78 \pm 4$ °W and has a magnitude of $0.52 \pm 0.15$ deg/Ma, which is in the same direction as predicted by ICE-6G_C, though with a smaller amplitude (about 40% less).

As discussed in the methods section, our approach also allows to quantify the contribution to GRACE from mass changes in the Earth's surface water layer. Results are shown in Fig. 2.

The largest signals can be found over the two ice sheets and largely saturate the colour scale. In addition, some isolated glacier regions are evidently losing mass, such as Alaska and Patagonia, while those neighbouring Greenland, such as the Canadian Arctic and Iceland, are not directly distinguishable due to the low resolution of the geoid representation. A few main regions

of large land hydrological variation are also evident, such as the mass loss in the Caspian Sea and the Northern India Plains, and the mass gain over the Zambezi River basin. The uncertainty is overall rather small, with the exception of some regions in North America, especially south of Hudson Bay and over Baffin Island.

At large scales, the geoid rates are dominated by a large positive signal at low latitudes and by a diffused negative signal in polar areas, mostly reflecting the global impact of polar ice melt on the oblateness of the Earth and on the position of its rotation

axis.

In Fig. 3, we show the reconstructed signal (GIA + water layer) and its residual with respect to the original GRACE trend. Particularly interesting is the plot of the residuals, in the bottom panel: apart from the clear signature of the 2004 Sumatra-Andaman and of the 2011 Tohoku-Oki megathrust earthquakes, which we expressly do not model, most of the remaining

signals are at least one order of magnitude smaller than those in the reconstruction shown in the top panel. In the regions characterised by the largest signals, the residuals are minimal, indicating that the chosen fingerprints are adequate to reproduce the input GRACE fields.

At global scales, the main residual signal is represented by a positive band between about 20 °N and 40 °S, with peak values in the SE Pacific and in the Indian Ocean, likely due to the combined effect of the inaccuracy of GRACE in determining

changes in the Earth's oblateness and additional noise, or unmodeled mass redistribution, at other low frequencies.

From the results that contribute to Fig. 2, we can estimate global mean ocean mass changes due to individual sources. Those values are listed in Table 1, and they are especially meant for validation of the ensemble solution. At the same time, the

uncertainties provide an indication of the role of GIA in GRACE-based estimates. The values and uncertainties for terrestrial water storage (TWS), GIA, and global ocean are obtained from integrating the individual signals over all oceans, after converting geoid changes into equivalent-water-height changes (Wahr et al., 1998), and excluding a 300-km-wide coastal buffer zone. The values for the three ice sources are obtained from direct scaling of the original fingerprints, which avoids possible biases from near-field sea-level changes (Sterenborg et al., 2013). We are comparing results against Frederikse et al. (2019) and Bamber et al. (2018), in the following indicated as F19 and B18, respectively. The global ocean mass change is considerably smaller than the estimates by F19, mostly due to more negative values from TWS, though with a large uncertainty. The contributions of the individual ice sources are very close to the results by B18, where the largest term is Greenland, followed by glaciers and Antarctica. We estimate a GIA contribution to global ocean mass change of $0.8 \pm 0.5$ mm/yr, whereas Tamisiea (2011) estimated values between 0.8-1.7 mm/yr for the same quantity.

Finally, in Table 2, we list estimated trends of geocentre motion, Earth's dynamic oblateness, and secular polar motion. Geocentre motion is only provided for the water layer, since there are no benchmarked solutions we can rely upon to generate the corresponding GIA fingerprints. Our results for geocentre motion are consistent with the estimates of Sun et al. (2016b) and Rietbroek et al. (2012).

As far as $J_2$ is concerned, the total value of $3.5 \pm 1.1$ e-11/yr is in line with some solutions based on satellite laser ranging (Sośnica et al., 2014) as well as to one of our previous solutions based on the GRACE-OBP approach (Sun et al., 2016a), though with respect to the latter it results from smaller individual contributions by GIA and the water-layer.

About secular polar motion, it is interesting to notice how the component along the Greenwich meridian, represented by the spherical harmonic coefficient C21, is the largest component and almost entirely due to mass transport in the water layer, while the perpendicular component (S21) is mostly due to GIA. When added together, the GIA and water layer contributions are able to exactly reproduce the direction of the secular polar motion observed by GRACE (17 °W) as well as 95% of its magnitude (GRACE: 1.26 deg/Ma).

## 4    Discussion

The core of the proposed approach, and its main innovation with respect to the original work by Rietbroek et al. (2012), lies in the expectation that the used set of fingerprints is sufficiently orthogonal to allow for a unique solution of the problem based on a single set of observations, i.e., GRACE data. We expect this to be the case for signals due to mass redistribution within the water layer, since the sources are small and sufficiently separated in the spatial domain. The problem becomes more complex when mass change sources overlap in the spatial domain: in this case, the use of a single dataset could fail to provide a unique solution. In particular, we are not able to solve for co-located GIA fingerprints, such as those that would result from varying mantle viscosity for a given ice history. For this reason, we are producing different sets of GIA solutions, each based on a single combination of ice histories and mantle viscosity, and average them afterwards. Similarly, we are not able to use

smaller GIA patches over the ice sheets, since their scale would become comparable to that of present-day ice mass changes. Hence the choice of using a single GIA fingerprint for Antarctica and no GIA fingerprint at all for Greenland. Working with overlapping signals would require the use of additional datasets and/or regularization methods (e.g., Wu et al, 2010; Rietbroek et al., 2016). In this study, we have chosen to adopt the simplest possible approach, by using only one dataset and no additional regularization, with the aim of providing a robust solution in terms of internal consistency and of global mass conservation, and of maintaining control on the impact of input data on the final solution.

In order to prove the orthogonality of the fingerprints and the uniqueness of our inversion, in Fig. 4 we show the correlation matrix of the individual fingerprints. In particular, the seven GIA fingerprints are displayed in the top and right end of the matrix and show a generally low correlation with the rest. The most serious problems would be expected over Antarctica, due to the overlap between GIA and present-day ice mass change: however, the Antarctic fingerprint (gia_001, seventh line from the top) is only showing a correlation larger than 0.8 with ant_001, which represents that part of the WAIS draining into the Ronne Ice Shelf, and between 0.8 and 0.6 with ant_017, along Siple Coast (WAIS, draining into the Ross Ice Shelf). Since the present-day mass change of Antarctica is represented by 25 drainage basins and 10 peripheral glacier regions, and the largest mass loss is not coming from those highly correlated regions, we think we can consider the inversion over Antarctica to be a well-posed problem. Concerning other regions, we only see some larger correlations over Greenland, mostly concerning high and low elevation sector of the same basins, and over a few adjacent glacier regions. Those likely reflect a limit in the capability of GRACE of resolving such concentrated signals, but they do not represent a problem for the estimation of large-scale mass changes.

Additionally, in Fig. 5, we compare our GIA solution against a few other global models used by the geodetic community: A et al. (2013), Peltier et al. (2015), and Caron et al. (2018). Note that, since most GRACE-based studies are concerned with surface mass redistribution, we represent the differences in terms of equivalent water height trends. The model by A et al. (2013) is based on the ICE-5G ice history (Peltier, 2004), which was superseded by ICE-6G_C in Peltier et al. (2015), while Caron et al. (2018) is based on a scaled version of the ANU ice history (Lambeck et al., 2010, 2014) for the Northern Hemisphere and of IJ05 (Ivins&James, 2004) for Antarctica. Over Antarctica and Northern Europe, including the Barents Sea, our ensemble solution is slightly smaller, but rather similar to Caron et al. (2013): this could be expected, since both models are based on the same initial ice histories. Over North America, our results are closer to Peltier et al. (2015), though our predicted surface mass change signal is generally smaller over and around Hudson Bay; concerning the comparison with the other two models, a large residual can be found west of Hudson Bay with respect to both of them, and south-east of Hudson Bay with respect to Caron et al. (2018). Over Greenland, the differences with the other models are due to the fact that have chosen not to use any GIA fingerprint for this region, as explained earlier.

One of the interesting applications of GIA models to GRACE estimates of present-day surface mass redistribution concerns the quantification of the TWS contribution to global mean sea level change, which is still very uncertain. A recent paper by Jensen et al. (2019) estimates it by using the GRACE product ITSG-Grace2018s and the GIA model by A et al. (2013). In their Fig. 1, a large region of mass loss can be seen west of Hudson Bay, with peak values of more than -20 mm/yr e.w.h.. In

approximatively the same area, our ensemble GIA solution is much smaller, with a maximum difference of about 30 mm/yr (Fig. 5b): we argue that the large mass loss signal of Jensen et al. (2019) west of Hudson Bay could be an artefact caused by mismodelled GIA.

## 5    Conclusions

We have partitioned GRACE monthly fields into a linear GIA contribution and the time-varying effect of the redistribution of
230 water masses at the Earth's surface, and then computed a linear trend of the latter. The fact that the residual between the original GRACE trend and the sum of GIA and water redistribution trends does not show any large signal gives us confidence that the proposed fingerprint approach is capable of reproducing the effect of the different physical processes at play. In addition, the contributions of individual sources to global mean ocean mass change are in line with the most recent literature, while their uncertainties are meant to provide a realistic quantification of the global role of GIA in GRACE-based estimates
of present-day water mass redistribution.

In the future, we expect to improve the spatial resolution of our empirical GIA model, thanks to the longer time series provided by the GRACE Follow-On mission, and to a more advanced treatment of observational noise.

### Data availability

The data used to generate Fig. 1-2, as well as a spectral representation of the ensemble GIA solution and of the various
components to the trend in surface water redistribution, is publicly available through the 4TU.Centre for Research Data at https://10.4121/uuid:4ecc3333-a25b-477a-a373-0503423ec9b1. Supporting datasets, such as monthly reconstructions of surface water redistribution, are available upon request.

### Author contribution

Y.S. and R.R. devised the study and analysed the results; Y.S. performed the calculations, produced the figures and commented
on the paper; R.R. wrote the paper.

### Competing interests

The authors declare no competing interests.

## Acknowledgements

We thank Lev Tarasov, Anthony Purcell, and Dick Peltier for making available their ice sheet history reconstructions; Pavel Ditmar and Roelof Rietbroek for discussions on an early version of this study; John Ries and Mark Tamisiea for discussing the meaning of the GRACE pole tide; Matt King for commenting on geocentre motion. Y.S. is supported by the National Natural Science Foundation of China (grant 41801393), the Education Department of Fujian Province (grant JT180031), and Central Guide Local Science and Technology Development Project (grant 2017L3012). He is also partially supported by the QiShan program of Fuzhou University. R.R. acknowledges funding from the Netherlands Organisation for Scientific Research (NWO), through VIDI grant 864.12.012.

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

| | GMSL (mm/yr) | Error (90%) | Mass (Gt/yr) |
|---|---|---|---|
| **Global ocean** | **1.24 [ 1.74 / - ]** | **0.38** | **-450 ± 140** |
| Glaciers | 0.52 [ 0.63 / 0.55 ] | 0.03 | -190 ± 10 |
| Greenland | 0.68 [ 0.80 / 0.71 ] | 0.03 | -250 ± 10 |
| Antarctica | 0.33 [ 0.37 / 0.31 ] | 0.02 | -120 ± 10 |
| TWS | -0.29 [-0.06 / - ] | 0.36 | 110 ± 130 |
| **GIA** | **-0.80** | **0.81** | **290 ± 290** |
| | | | |

**Table 1. Estimated global mean ocean changes (Jan 2003 – Aug 2016), in terms of global mean sea level (GMSL) and mass, for the global ocean and its individual contributors. Between brackets: values from Frederikse et al. (2019) and Bamber et al. (2018), respectively. Estimates from Bamber et al. (2018) are obtained from their Table 2, by averaging results over the three consecutive time windows covering the GRACE era. We assume that 1 mm GMSL = -362 Gt, and we round off the mass estimates to the nearest ten.**

| | GIA | Water layer |
|---|---|---|
| X geocentre (mm/yr) | - | -0.05 ± 0.03 |
| Y geocentre (mm/yr) | - | 0.14 ± 0.05 |
| Z geocentre (mm/yr) | - | -0.35 ± 0.08 |
| $J_2$ (1e-11/yr) | -2.5 ± 0.9 | 6.0 ± 0.6 |
| C21 (1e-11/yr) | -0.16 ± 0.08 | -1.52 ± 0.18 |
| S21 (1e-11/yr) | 0.74 ± 0.33 | -0.24 ± 0.31 |

**Table 2. Estimated linear trends (Jan 2003 – Aug 2016) of geocentre motion, Earth's dynamic oblateness ($J_2$), and secular polar motion (C21, S21). Errors represent the 90% confidence level. C21 and S21 can be converted into units of deg/Ma by multiplying them by a factor of -6.9e10 (Chambers et al., 2010).**

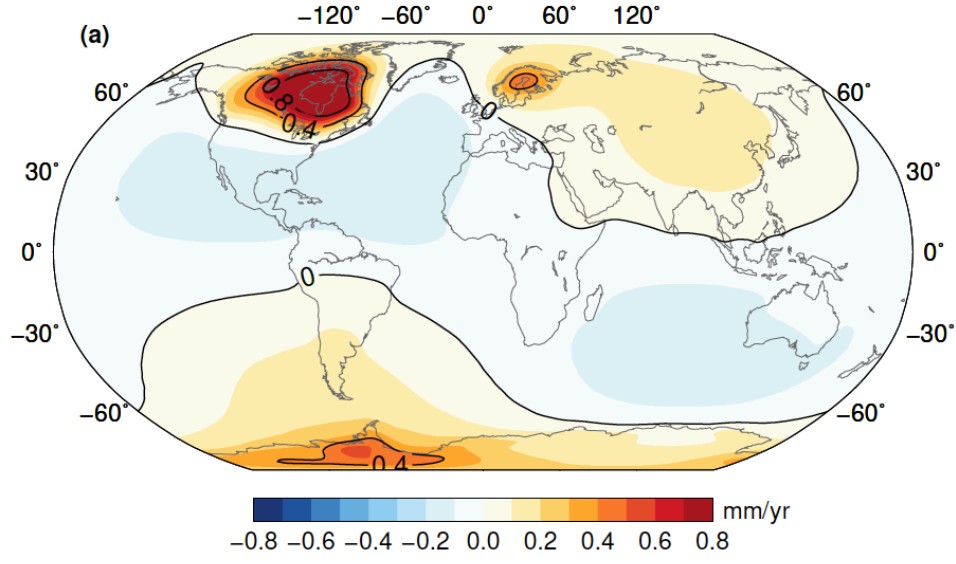

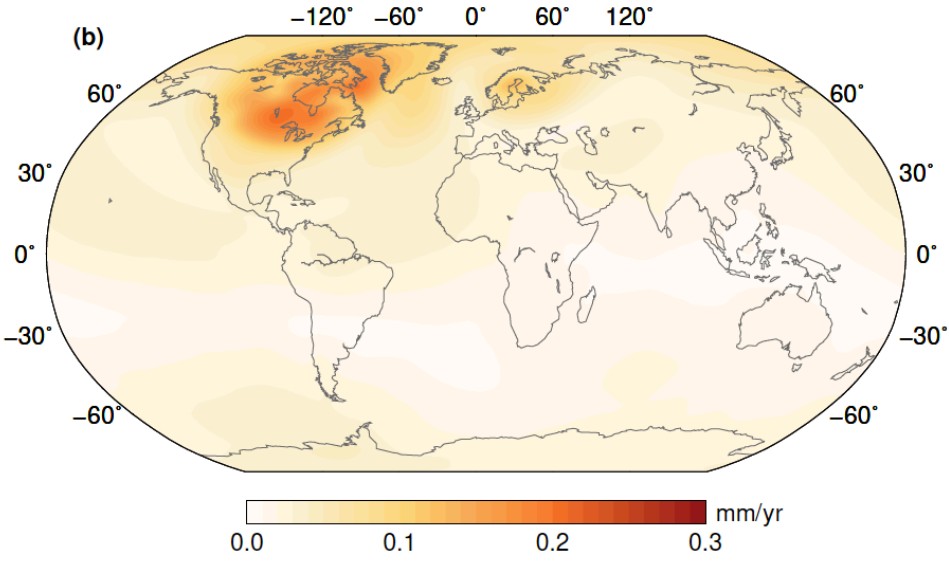

**Figure 1. Ensemble GIA solution (top), and one standard deviation (bottom), in terms of geoid height trend.**

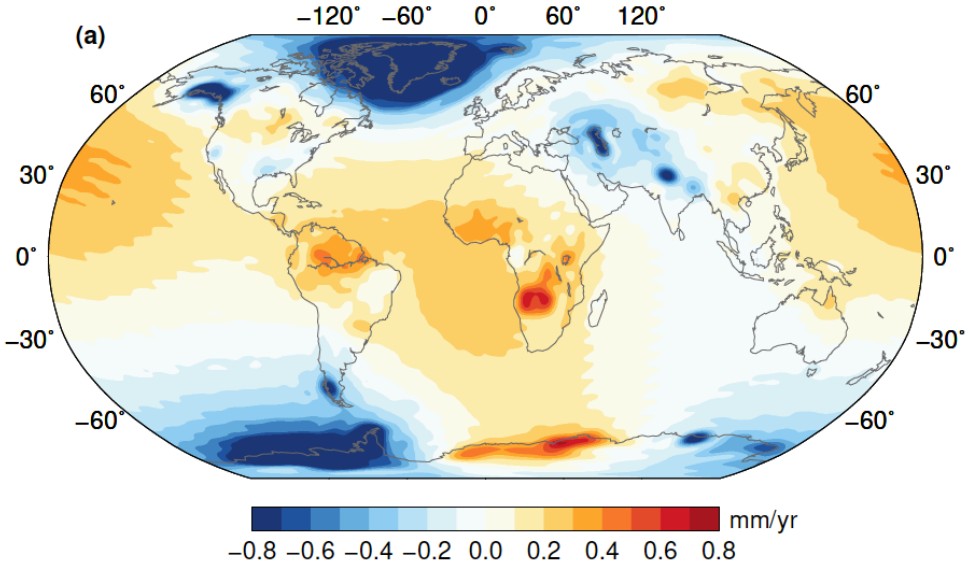

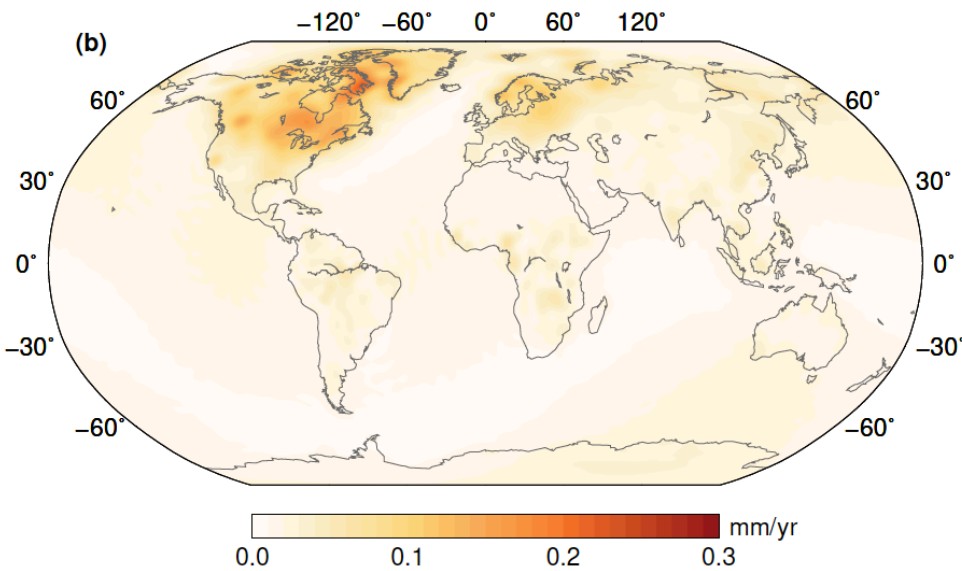

**Figure 2. Ensemble solution for the effect of mass redistribution in the water layer (top), and one standard deviation (bottom), in terms of geoid height trend.**


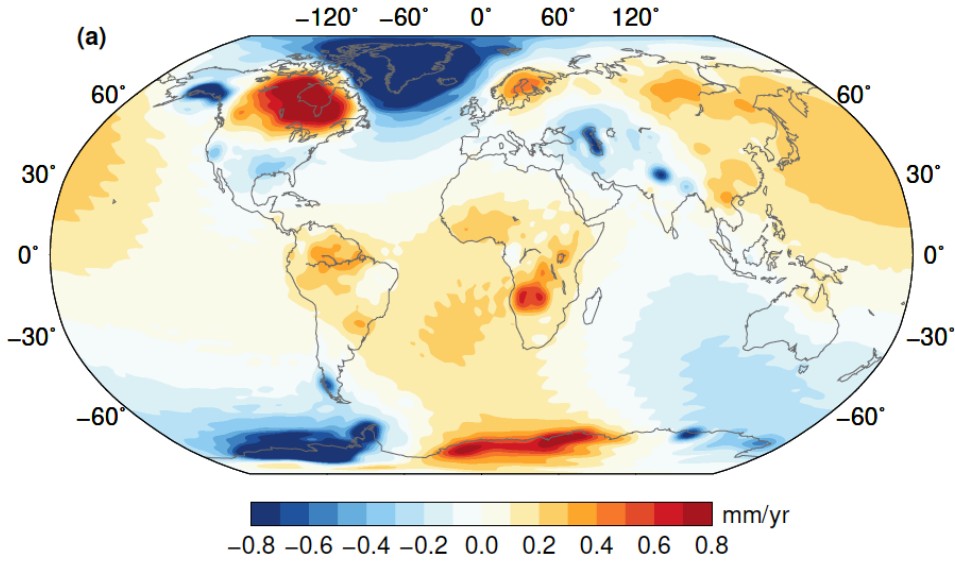

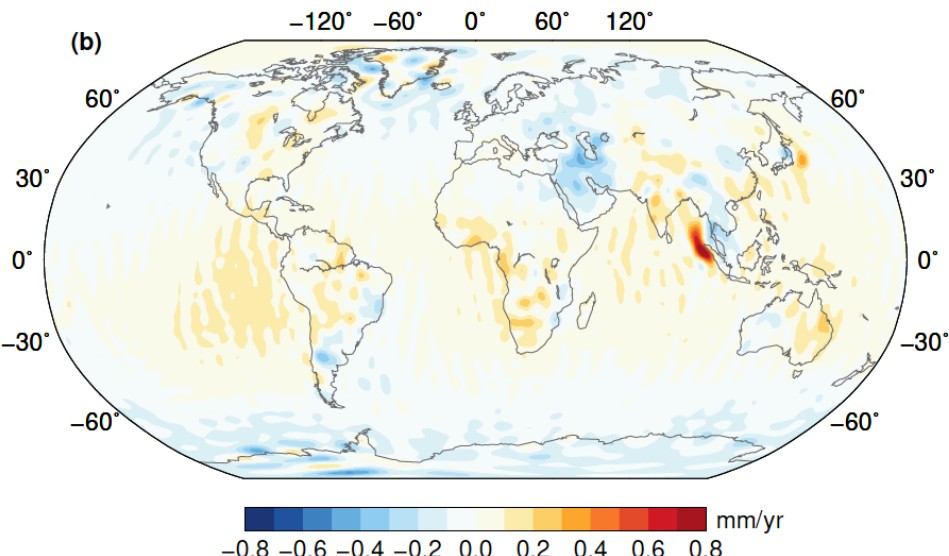

**Figure 3. Ensemble GRACE reconstruction (top), and residual GRACE signal (bottom), in terms of geoid height trend.**

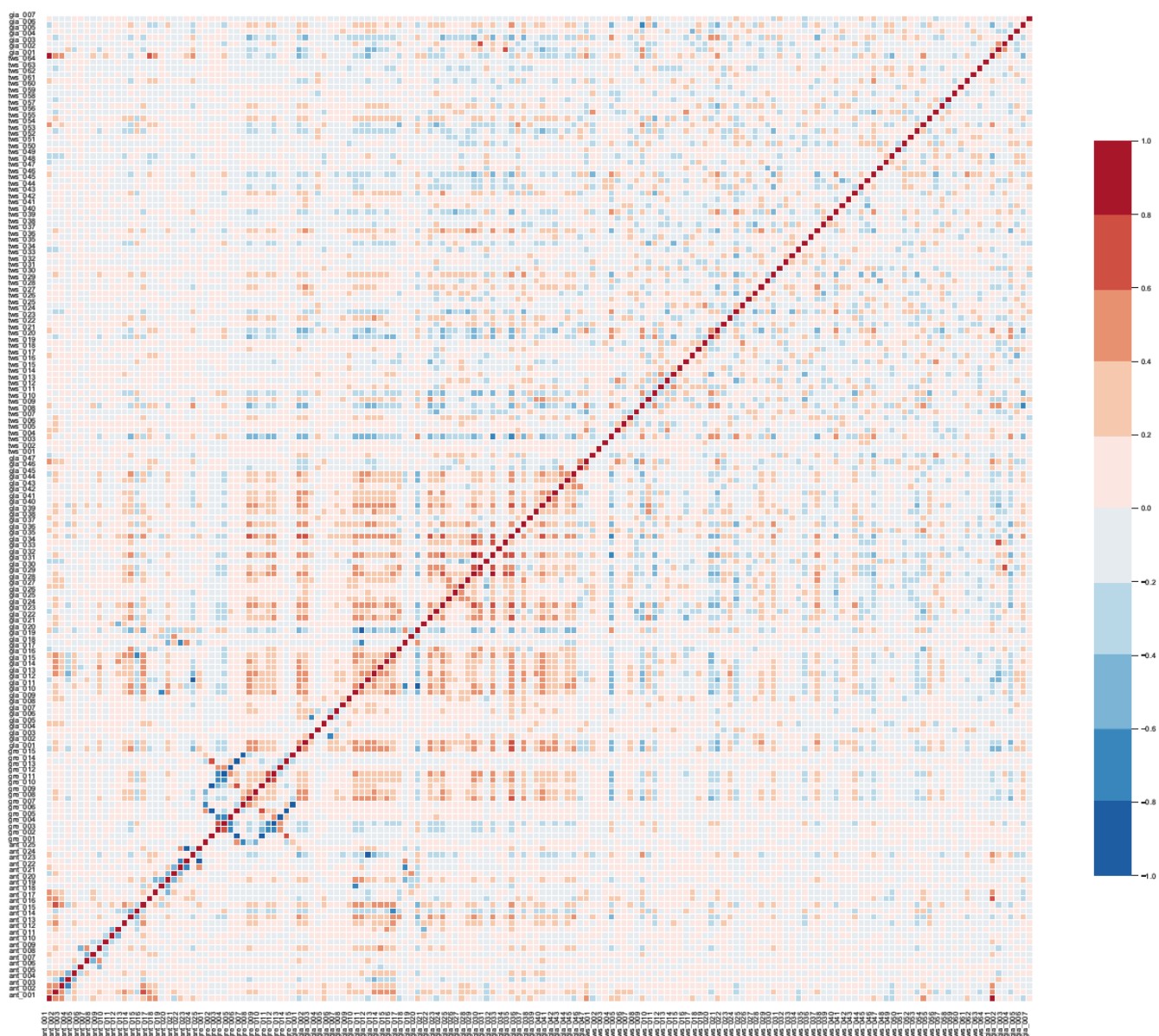

Figure 4. Correlation matrix of the 158 fingerprints used in this study. The GIA fingerprints show Antarctica (001), Northern Europe (002-003), North America (004-006), and the secular pole tide contribution (007). The location of the sources of the individual fingerprints for ice sheets and glaciers can be found in the online supplement of Sun et al. (2019).

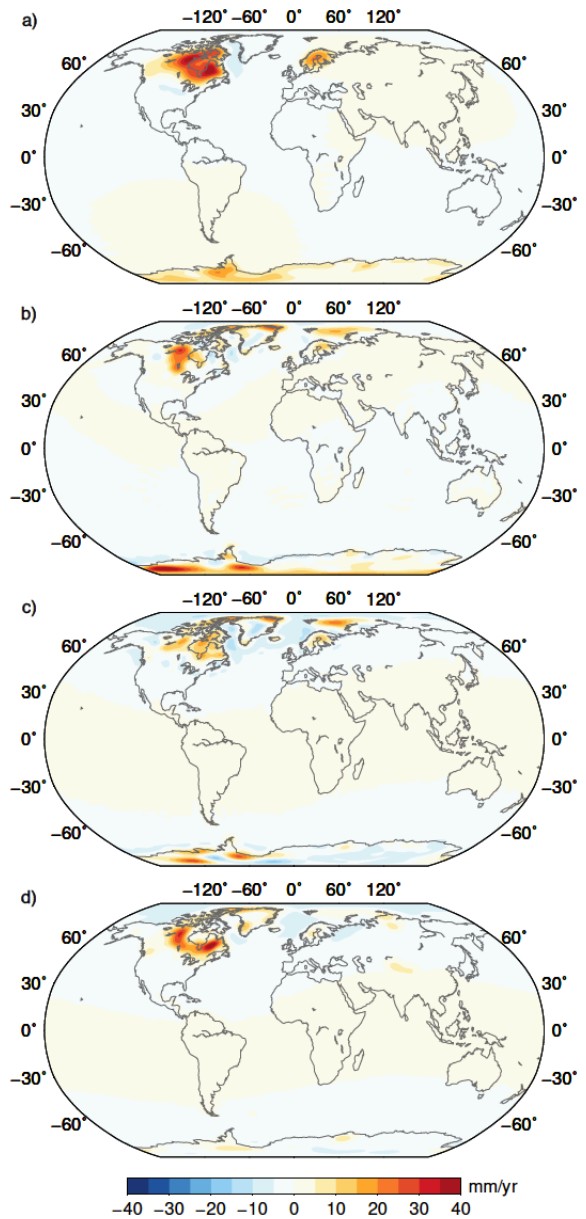


**Figure 5. a, Ensemble GIA solution in terms of equivalent water height; b-d, difference between three published GIA models and the ensemble solution (mm/yr e.w.h.): b, A et al. (2013) – ensemble; c, Peltier et al. (2015) – ensemble; d, Caron et al. (2018) – ensemble.**
