# Peer review of "A global semi-empirical GIA model based on GRACE data"

_Earth System Dynamics, 2019_

## Referee Comment (RC1) · Don Chambers (Referee) · 30 Sep 2019

This paper describes a dataset that includes GIA maps and water mass anomaly trends that are based on fitting pre-computed fingerprints to GRACE observations. This is a modification of work the authors and Reitbroek et al have previously done.

Overall, I think this is a well-written manuscript and the results look very exciting. I think the idea of fitting the fingerprints to just GRACE observations and not adding in complexity of non-homogeneous GPS and altimetry, steric, and climate models (as in reitbroek et al.) is superior as less prone to contamination from errors in the other datasets.

One issue, which the authors do acknowledge, is the assumption that their basis func-

tions (the fingerprints) are orthogonal. If they aren't, then the estimated scaling parameters will be correlated and they can't really be treated as independent. This isn't really a problem if they cluster regionally and are independent of other regional clusters. The problem in this solution may be that the GIA over Antarctica may be correlated with the mass loss over Antarctica, so they can't really be treated independently.

The authors have all the information needed to test the level of correlation in their covariance matrix from the least squares estimation. I would like to see some analysis of the correlations between estimated parameters that can be computed from this matrix – in particular, the correlation between the GIA parameters and the Antarctica parameters. If these are NOT correlated significantly, then great – the authors have demonstrated that their GIA model can probably be treated independently from their mass loss over Antarctica. If the they ARE correlated significantly, then they need to make some cautionary statements acknowledging this.

My only other comment would be a request for the authors to also include their GIA patterns and other patterns in terms of gravitational spherical harmonics (and not just geoid rates). This will allow easier combination for anyone using GRACE data to convert to water storage – this isn't the same as geoid height.

Other than these two minor issues, the paper is really nice and I encourage publication if the authors will comment on the correlation issue based on their covariance matrix.

---

## Referee Comment (RC2) · Anonymous Referee #2 · 8 Oct 2019

This submitted research report builds upon work published by Yu Sun et al (2016 a, b; 2019) which developed a fingerprint- empirical orthogonal functional analysis of GRACE level 2 products and demonstrated their utility in dealing with terrestrial water storage and cryospheric change over the time scale of the GRACE mission. These papers were both valuable and intellectually stimulating, and also treated glacial isostatic adjustment (GIA), though without certain nuances that we see in the current submission.

The current submission attempts to isolate a new empirical GIA model by developing an ensemble based upon forward modeling, finding some statistics of those models and then by addition/subtraction from the GRACE-OBP methodology of Sun et al. (2019, GRL 46 https://doi.org/10.1029/2018GL080607) deliver an empirical model

based upon GRACE RL06 alone. This is a worthy goal, but I am unconvinced that the results are valuable enough for a publication at this time.

One of the greatest difficulties that the authors are faced with is that of properly dealing with, and quantifying, the error propagation, and they do not appear to have dealt with this in any way. Furthermore, the ensemble forward model set has little to offer that convinces me that it is a statistical sampling. More damning is the fact that the authors don't seem particularly convinced themselves: A careful reading of the Conclusions (section 6) is in order. The first paragraph states: "In addition, the estimated ocean mass change and the contributions of its individual sources are in line with most of the recent literature". And in the second paragraph they provide equally unenthusiastic statements about GIA: " The uncertainties obtained for the individual contributors . . . provide a realistic quantification of the global role of GIA in GRACE-based estimates of present-day mass redistribution." This statement is tantamount to an admission of failure. My rationale is as follows. The authors set out to find an empirical GIA model. But the model they find has uncertainties as large as the uncertainties in determining GRACE-based mass changes. When I first did a skim reading of this submission, my impression was that this is a small epsilon forward with GRACE-GIA modeling. Upon digesting the paper for review my view changed to thinking that the sign on the epsilon is, in fact, negative.

In light of the fact that the paper has now gotten to the Discussion phase, let me below try to be helpful (yet, none-the-less, quite critical) in some more detailed comments/observations. Perhaps the submission can be resuscitated after a major rewriting and resubmitting with a less 'assertive' title.

Details.

1st sentence Abstract. This statement is generally false. At best, GIA models preform updates between viscosity models and ice sheet history models. It is very rare that they are solved for simultaneously, and especially if the ice flow is computed dynamically.

[Figure]

In the abstract, it would be nice to know what the 1 or 2 sigma uncertainty is in the J2 dot solution using the GRACE-OBP method and that which is GIA-empirically determined.

Line 30, Section 1. The referred papers for earth rotational GIA are more that 30 years old. The modern literature is actually quite rich, and I suggest referencing a paper like those published by Nakada or Mitrovica during the last 5 years.

Line 46, Section 1. The approach is claimed to be like one developed originally by Rietbroeck. But the latter used ocean altimetry, and that is not being used here. The authors need to clarify. Would the current study have been more successful if also employing ocean altimetry, or less so?

Line 72, Section 2.1. The use of the phrasing "allows to" should be changed as it is grammatically incorrect. Use something like, "provides sufficient information for partitioning".

Lines 72-73, Section 2.1 The use of "somehow" is quite odd in this context. (This occurs later). This is a red flag for a reviewer. If it is a 'somehow' then there is something that the reader must take as either suspicious or done with uncertainty, at best. It therefore needs much clarification.

Lines 80 – 85 Section 2.2. As described, this is not an ensemble. No real statistics can be derived from these 4 forward models. One could say: The spread is ... and we assert that it is representative of the one sigma about some average. But there is utterly no statistical significance to these. And, by the way, each of the models have their own mountains of data that both support and bias them, and that have data gaps and under/oversampling, none of which is being treated in your empirical procedures.

Line 89, Section 2.2. "think" -> "thick"

Lines 104-108, Section 2.3. This entire explanation of the procedure to deal with rotationally related surface mass change and GIA has to be redone. I suggest writing out

the terms in equation form: What exactly in the end is solved for, and what is the uncertainty? I am completely lost, and if I read the logic – verbatim – I would be led to the conclusion that the paper is simply wrong. I have faith that the author's understanding of all this is more-or-less sound, so I think it is just a series of poor explanation(s) that is involved. It's really a weak part of the paper, however.

Lines 109-111, Section 2.2. This statement seems contradictory. If not included, why discuss it here and later in the paper? I am quite confused.

Lines 125-129, Section 3. Again, we return to this confusion about pole-tide and its appearance in the fingerprint maps. What does this mean, or does it even have a meaning? Is it perhaps an artifact? I am not criticizing, I am just confused.

Lines 137-139, Section 3. The sentence starting "The uncertainty overall . . . " is disturbing, for this is exactly where an empirical model, if at all valuable, might help advancing science. I return to this criticism below, with respect to the 'coupling' to hydrological signal.

Lines 149-153. Section 3. The proposed explanation for the round region in south Atlantic seems very speculative: a trend in TWS not removed in the pole-tide correction. (?) There should be more explanation to justify this. Why not cumulative errors propagated from GRACE RL06, including the de-aliasing models? Since no error propagation analysis was conducted here, then how do we assess this assertion?

Lines 155-156. Section 3. "At the same time, the uncertainties provide . . .". I suggest the same comment that was mentioned concerning the vagaries of the conclusions. Now those apply here as well. I elaborate on the criticism. I refer to a recent paper by Jensen, L., Eicker, A., Dobslaw, H., Stacke, T., & Humphrey, V. (2019). Long-term wetting and drying trends in land water storage derived from GRACE and CMIP5 models. J.Geophys. Res.: Atmospheres, 124. https://doi.org/10.1029/2018JD029989. In Figure 1 of this paper we see a map of the TWS trends from ITSG-Grace2018s. In that map, note the region of negative ($\sim$-10 to -20 mm/yr equivalent water) to the west

and south of Hudson Bay, and that also rings across northern Canada in red, and the outer blue ring at amplitude 5 – 15 mm/yr outside of these. Are those really TWS, or are they mismodeled GIA? We can deduce that such questions should apply just by looking at the Jensen 2019 map. So, what, if anything, has the current paper submitted to ESD contributed to this? Has it (the present paper) just affirmed what we surmise from Figure 1 of Jensen?

Lines 158-160, Section 3. A 300 km buffer is employed. But no Gauss filter is mentioned, so I assume it is not used, and all the signal generated by fingerprint EOFs. What does the buffering do? Some of that buffering will get rid of real signal, not just gravitational artifact, as was the minor point being made by Sterenborg.

Lines 165-167, Section 3. The estimate of GIA signal is compared to that of Tamisiea. But this is difficult to make much of, as the statement is too equivocal: It is better to state it quantitatively: "We estimate a GIA [thing] of x.x $\pm$ y.y, whereas Tamisiea estimated b.b $\pm$ c.c, for that same GIA [thing]."

Lines 175-178, Section 3. A lot is made here of determining a partition of the J2 signal, and convincing (and seemingly rigorous) work was established in Sun et al (2019). But this disambiguation, as reported here, seems notably unconvincing without evaluating error propagation, or at minimum, estimation.

Lines 190-192, Section 4. The statement: "... by using only one dataset we get ... on the final solution", is a good one, and maybe in a Brevia paper to this journal that explicit point can be made, even convincing some that it is important! But it is quite challenging to recast this work into something that would convince us that science is being advanced, even by a small epsilon.

Lines 201-207, Section 5 and final remarks. Again, the big deficit to this paper is that lack of any attention to error propagation, as I suspect that if that were done a similar, but quite useful quantitative conclusion might be discovered. Such quantification could become a valuable thing, especially with respect to planning the next generation of

space gravimetry missions.

---

## Referee Comment (RC3) · Lambert Caron (Referee) · 15 Oct 2019

"A global empirical GIA model based on GRACE data" by Yu Sun and Riccardo E. M. Riva presents a joint least-square inversion on a global scale of GIA and contemporary surface water mass redistribution solely constrained by GRACE data. In addition to providing a spatial description of both components, the authors also provide estimations of their respective contribution to global mean sea level change, changes to the Earth oblateness coefficient and geocenter motion. The approach is built on solid theory and combines all of the important sources of signal to analyze the signal. In addition to enriching the spectrum of GIA models available to correct GRACE data, this work has the potential to bring new and valuable information for the GIA modeling community by highlighting the ability of GRACE to constrain GIA signal, and to point out

how the resulting solutions would place themselves among or differ from the existing GIA models. Unfortunately, the present version of the manuscript shows very limited comparisons with other GIA models and the uncertainty quantification scheme is not sufficiently explained for the reader to assess (one way or the other) the robustness of the inversion and the degree of independence between all of the fingerprints used. I think the impact of the paper would be much greater if these points were elaborated on and the statistical analysis was shown more explicitly to be both rigorous and supporting the conclusions robustly. I believe the authors should already have all of the right statistical tools at their disposal to answer these questions. Provided the authors address these points in a subsequent revision I would certainly support the publication of this manuscript.

Major points: -l39: I know that other authors have used these terms in previous papers in a somewhat interchangeable manner, but I think it is important to distinguish data-driven from empirical. GIA models derived from partial differential equations (e.g. using love numbers) are not empirical (they are based on a physical theory) but they can be data-driven if their parameters are inverted from a dataset. Among such models are for example Peltier et al. (2015), Lambeck et al. (2014) or Caron et al (2018). Because the authors use such theory to generate their fingerprints, I would argue that their approach is not empirical (and I believe the title should reflect that), and in fact amounts to rescaling the loading history via the least square coefficients as was done in the aforementioned papers, and others before them. In my opinion, that is something the authors could put forward as an advantage of their approach, as it means it is consistent with how we otherwise model and understand the physics behind surface loading and deformation of the Earth interior.

In particular, it means we could compare the GIA scaling coefficients (here the inverted coefficients of the fingerprints) with the values found in the literature and that are based on inverting RSL, and other datasets. That exercise cannot easily be done with true empirical models as they are not built on comparable basis functions. An

important question this paper could (begin to) illuminate by showing these coefficients is therefore: are GIA models preferred by GRACE statistically different from the ones constrained with traditional datasets?

-l81: What is the impact of the number of evaluated cases (here 4) on this statistical analysis? Would the authors expect a lot of differences from a more comprehensive exploration of the parameter space (particularly the viscosity profile)? How much does this limit the applicability of these results to correct GRACE?

-l88: Ice histories such as that of ANU and ICE-6G_C have been crafted such that when combining all of their regional components, they are able to explain paleo RSL data (especially through the eustatic sea level curve). By recombining regional components of different models, is the solution still consistent with what we know about past RSL - and therefore part of what validates these ice histories in the first place?

-l113: The authors unfortunately do not really elaborate on their uncertainty quantification approach, and only state that they combine all 4 solutions into an average. How did the author calculate their standard deviation map? Did they: a) take the least-square optimized signal of each of the 4 cases, and then calculated the standard deviation between them (which the first sentence at l119 seems to point to), b) calculated the variance/covariance matrix of the coefficients for each case from the least-square system, which using the notations of Yun et al. (2019) should be a term with a form along the lines of (F'T'PTF)^-1, and then averaged that covariance matrix between the 4 cases, c) a method similar to b), with a weight associated with each of the 4 cases in the averaging process to take into account that some of them allow smaller residuals than others, d) use yet another method?

Out of these possibilities, a) is not an appropriate estimator, it would underestimate the uncertainty as it neglects the level of constraint of each least-square inversion. One could imagine a situation where all 4 best fit produce a similar signal for a given grid point or Stokes coefficient, but with a high variance/low confidence for that value. b)

assumes that all 4 cases should have the same weight, which would be acceptable if they yield a similar sum of the residuals, c) being be more indicated otherwise. As this explanation is missing, it is difficult for me to understand and critically examine the results section of the manuscript, and going back to Yun et al. (2019) which details the method, I could not find the information related to uncertainty quantification either. I would add that if the authors mean to provide their model to the GRACE community for correcting GIA, it is very important that the treatment of uncertainty quantification be transparent. -l179-182: An additional benefit of showing the covariance matrix of the least-square coefficients is that one can verify the degree of independence (or some measure of it, at least) between the different fingerprints by transforming it into a correlation matrix. This way sufficient orthogonality does not have to remain an assumption.

Minor points: -l8: if the authors are referring to RSL indicators, they only point to a local level, not global

-l35: This reference should be Caron et al. 2018, not 2017

-l78: Why do the authors assume the Earth to be compressible for the fingerprints of the previous section but incompressible for GIA deformation? Is this not inconsistent?

-l81: It was not clear for me at first read whether the authors were combining ICE-6G in one region with another model in the other region, despite the previous sentence. I suggest rewording along the lines of: "we use either GLAC1D (Tarasov et al. 2012) and ANU (Lambeck et al. 2010) in North America and Northern Europe, respectively, or ICE-6G_C (Peltier et al. 2015) in both regions."

-l81: "of" should read "or"

-l88: The authors reference Ivins & James (2005) for the IJ05 model, which had an updated version (IJ05_R2) released in 2013 (Ivins, E. R., T. S. James, J. Wahr, O. Schrama, J. Ernst, F. W. Landerer, and K. M. Simon (2013), Antarctic contribution

to sea level rise observed by GRACE with improved GIA correction, Journal of Geophysical Research: Solid Earth, 118(6), 3126–3141). If the authors used the updated version, this is simply a matter of updating the reference, but if not I would be curious to know why they chose the old version and if they expect a significant change from this choice. The volume of the Antarctic ice sheet at the LGM is different by about a factor 2 for example.

-l116: "rounder": do you mean smoother or with a more circular shape?

---

## Author Comment (AC1) · 13 Dec 2019

Response to comments by Referee 1 (Don Chambers).

We are thankful to Dr. Chambers for his nice words about our study. We were delighted to read that he has particularly appreciated our idea to fitting the fingerprints to a single dataset: a choice that was indeed made to avoid possible biases introduced by the combination of different techniques.

Comment 1
"*One issue, which the authors do acknowledge, is the assumption that their basis functions (the fingerprints) are orthogonal. If they aren't, then the estimated scaling parameters will be correlated and they can't really be treated as independent. This isn't really a problem if they cluster regionally and are independent of other regional clusters. The problem in this solution may be that the GIA over Antarctica may be correlated with the mass loss over Antarctica, so they can't really be treated independently.*
*The authors have all the information needed to test the level of correlation in their covariance matrix from the least squares estimation. I would like to see some analysis of the correlations between estimated parameters that can be computed from this matrix – in particular, the correlation between the GIA parameters and the Antarctica parameters. If these are NOT correlated significantly, then great – the authors have demonstrated that their GIA model can probably be treated independently from their mass loss over Antarctica. If the they ARE correlated significantly, then they need to make some cautionary statements acknowledging this*."

As suggested, we have computed the correlation of the covariance matrix, which we show below (Figure R1) and which will be added to the paper. Since we make use of 158 fingerprints (from top to bottom and from right to left: 7 GIA, 64 TWS, 47 glaciers, 15 GIS, 25 AIS), the labels are somehow small. Antarctic GIA is represented by the fingerprint gia_001 (7th line from the top) and it is only showing a correlation larger than 0.8 with ant_001, which represents that part of the WAIS draining into the Ronne Ice Shelf, and larger than 0.6 with ant_017, along Siple Coast (WAIS, draining into the Ross Ice Shelf). A lower positive correlation, between 0.4 and 0.6, is also shown with basins ant_002 and ant_003 (EAIS, draining into the Filchner Ice Shelf), ant_018 along Siple Coast (WAIS, draining into the Ross Ice Shelf), and ant_024 over Graham Land (tip of the Antarctic Peninsula).
Since the present-day mass change of Antarctica is represented by 25 drainage basins and 10 peripheral glacier regions, and the largest mass loss is not coming from those correlated regions, we think we can consider the inversion over Antarctica to be a well-posed problem. Concerning other regions, we only see some larger correlations over Greenland, mostly concerning high and low elevation sector of the same basins, and over a few adjacent glacier regions. Those likely reflect a limit in the capability of GRACE of resolving such concentrated signals, but they do not represent a problem for the estimation of large-scale mass loss.

[Figure]

Figure R1: correlation of the covariance matrix.

Comment 2
*"My only other comment would be a request for the authors to also include their GIA patterns and other patterns in terms of gravitational spherical harmonics (and not just geoid rates). This will allow easier combination for anyone using GRACE data to convert to water storage – this isn't the same as geoid height."*

This seems to be a misunderstanding, since we had already published the GIA solution in terms of spherical harmonics, something that was possibly not very clear. While submitting a revised version of this manuscript, we will also submit spherical harmonics of the other trend solutions (total water layer, as well as four sub-components: Greenland, Antarctica, other glaciers and TWS).

Kind regards,
Riccardo Riva and Yu Sun

---

## Author Comment (AC2) · 13 Dec 2019

Response to comments by Referee 2 (anonymous).

We thank the Referee for their detailed comments about our manuscript. We have largely improved both the methodology and the estimation of the uncertainties. In combination with a number of additional explanations and rewordings, we believe we have been able to address all the points that have been raised.

Before a point-to point answer, we would like to explain the two major improvements.

I)      Error analysis.

Stimulated by the comments by Referees 2 and 3, we have decided to increase the number of viscosity profiles, in order to produce a more reasonable ensemble (indeed, "ensemble" was not a very appropriate term for the mean of four models).
In particular, we have produced GIA fingerprints based on all possible combinations of 6 viscosity values in the upper mantle (range 1e20-1e21 Pa s) and 11 viscosity values in the lower mantle (range 1e21-1e23 Pa s), giving rise to 66 different earth models. Each earth model has then been used in combination with two different ice histories (as before, ICE-6G or a combination of ANU+GLAC1D for the Northern Hemisphere; IJ05 for Antarctica in all cases), giving rise to 132 sets of GIA fingerprints. Each set is used independently, giving rise to 132 GIA solutions. The result that will be provided in the revised version of our manuscript will then represent the ensemble mean, and the standard deviation of this mean will represent the new solution uncertainty.
We note that the updated GIA model has a slightly smaller contribution to GMSL (-0.8 mm/yr instead of -0.9 mm/yr) and a larger uncertainty (0.8 mm/yr instead of 0.5 mm/yr, at 90% confidence).
More importantly, the spatial uncertainty patterns have become much more realistic, with generally larger values under the former ice sheets.
The results for $J_2^{\dot{}}$ reveal a very similar contribution from GIA ( -2.5 ± 0.9 1e-11/yr instead of -2.6 ± 0.2 1e-11/yr), and a 10% smaller contribution from the water layer (6.0 ± 0.6 1e-11/yr instead of 6.7 ± 0.1 1e-11/yr); the larger uncertainties are now more realistic.

Below, we reproduce a new figure showing uncertainties in the GIA model.

[Figure]

Figure R2: revised version of the bottom panel of Figure 1.

II)      Polar motion.

Some of the unclear wording used in the submitted manuscript was related to an effort to explain how we had dealt with the fact that a linear mean pole seemed to have been removed from the GRACE fields in the processing phase.

As it turns out, this was a misunderstanding from our side. In realty, that part of the linear mean polar motion that is due to GIA and to long-term changes in the water layer is still included in the Level-2 GRACE data used in this study (John Ries, personal communication). We have therefore modified the present-day fingerprints, in order to include the effect of the rotational feedback to the sea-level equation, which generated fingerprints with a much larger degree 2 order 1 coefficients.

In addition, we have improved how we construct the GIA degree 2 order 1 fingerprint (i.e., the GIA-induced polar motion fingerprint), by introducing a 2-step procedure. In step 1, we run the inversion by using six regional GIA fingerprints, generated without rotational feedback. We then take the six resulting pairs of degree 2 order 1 coefficients and we add them together to form a new GIA-induced polar motion fingerprint. In step 2, we run the inversion again, where the degree 2 order 1 coefficients of the six GIA fingerprints are set to zero, and where the GIA-induced polar motion fingerprint is treated separately (albeit with the C21/S21 ratio determined in step 1). In the original manuscript, we were directly building the GIA-induced polar motion fingerprint from the unscaled version of the six regional GIA fingerprints (i.e., step 2 only).

Both improvements together produced a GIA-induced polar motion solution with a considerably different direction (about 78° W instead of 88° W) and larger magnitude (0.52 deg/Ma instead of 0.37 deg/Ma). More importantly, when adding together the GIA and the water layer contributions, we are able to account for about 95% of the trend in both GRACE degree 2 order 1 coefficients.

Below, we reproduce the new figure showing the effect of mass redistribution in the water layer.

[Figure]

Figure R3: revised version of the top panel of Figure 2

Here follows a point-by-point answer to the Referee's comments (reproduced in italics).

1) *The current submission attempts to isolate a new empirical GIA model by developing an ensemble based upon forward modeling, finding some statistics of those models and then by addition/subtraction from the GRACE-OBP methodology of Sun et al. (2019, GRL 46 https://doi.org/10.1029/2018GL080607) deliver an empirical model based upon GRACE RL06 alone.*

We would like to clarify that there is no methodological link between the current study, which indeed builds upon Sun et al. (2019) and the GRACE-OBP approach discussed by Sun et al. (2016, JGR & JGeod), which represented an improvement upon Swenson et al. (2008).

2) *One of the greatest difficulties that the authors are faced with is that of properly dealing with, and quantifying, the error propagation, and they do not appear to have dealt with this in any way. Furthermore, the ensemble forward model set has little to offer that convinces me that it is a statistical sampling.*

See point I) above.

3) *More damning is the fact that the authors don't seem particularly convinced themselves: A careful reading of the Conclusions (section 6) is in order. The first paragraph states: "In addition, the estimated ocean mass change and the contributions of its individual sources are in line with most of the recent literature". And in the second paragraph they provide equally unenthusiastic statements about GIA: " The uncertainties obtained for the individual contributors provide a realistic*

*quantification of the global role of GIA in GRACE-based estimates of present-day mass redistribution." This statement is tantamount to an admission of failure.*

We regret having given the impression that we were not convinced by our own results. The fact that we managed to reproduced recently published estimates of ocean mass change was meant as an independent proof of the correctness of our results. In fact, the updated solution shows a smaller trend in ocean mass change (1.2 ± 0.4 mm/yr), mainly due to a larger negative contribution of TWS, in turn largely caused by properly accounting for polar motion. We consider this an important result of the current study, since the contribution of TWS to ocean mass change is still debated.
Concerning the second quoted sentence ("The uncertainties… mass redistribution"), we honestly do not understand how that could be tantamount to an admission of failure. If the issue is represented by the large GIA error (0.8 mm/yr at 90% level, with the new ensemble), we wish to strongly reject the strictly quantitative notion that the value of a model solely depends on its numerical accuracy.

4) *1st sentence Abstract. This statement is generally false. At best, GIA models preform updates between viscosity models and ice sheet history models. It is very rare that they are solved for simultaneously, and especially if the ice flow is computed dynamically.*

True. " that simultaneously solve for…" will be replaced by "that solve for both…".

5) *In the abstract, it would be nice to know what the 1 or 2 sigma uncertainty is in the J2 dot solution using the GRACE-OBP method and that which is GIA-empirically determined.*

The J2 dot uncertainties were listed in Table 2, but indeed they should have also appeared in the abstract, possibly specifying the individual GIA and water layer contributions. We note that now, with a larger ensemble, one standard deviation is 23% of the GIA contribution and 6% of the water layer contribution.

6) *Line 30, Section 1. The referred papers for earth rotational GIA are more that 30 years old. The modern literature is actually quite rich, and I suggest referencing a paper like those published by Nakada or Mitrovica during the last 5 years.*

We expressly cited the seminal papers, considering that the paragraph started with the word "Historically", but we will add references to a few recent papers, as suggested.

7) *Line 46, Section 1. The approach is claimed to be like one developed originally by Rietbroeck. But the latter used ocean altimetry, and that is not being used here. The authors need to clarify. Would the current study have been more successful if also employing ocean altimetry, or less so?*

Indeed, the present study makes use of a sub-set of the observations used by Rietbroek et al. (2012). However, this choice is deliberate, as discussed at the beginning of the discussion section (Sec.4) and highly appreciated by Referee 1. The problem with using different

datasets, especially when originating from independent satellite missions, is that it is extremely difficult to quantify possible systematic errors, such as those coming from representing all observations in a consistent reference frame. At this stage, we are not able to quantify the possible gain or loss of accuracy that the use of altimetry data would introduce, nor we think it should be the objective of this study, since it would require a considerable modification of our approach. Nonetheless, we will make clear in the introduction that one of the differences with respect to Rietbroek et al. (2012) is the use of a single data source.

8) *Line 72, Section 2.1. The use of the phrasing "allows to" should be changed as it is grammatically incorrect. Use something like, "provides sufficient information for partitioning".*

We will correct the sentence as suggested.

9) *Lines 72-73, Section 2.1 The use of "somehow" is quite odd in this context. (This occurs later). This is a red flag for a reviewer. If it is a 'somehow' then there is something that the reader must take as either suspicious or done with uncertainty, at best. It therefore needs much clarification.*

We agree with the comment, and we will remove the word "somehow" altogether, since the purpose was not to spread or hide doubts. We simply meant that the solid earth, the cryosphere and land hydrology are all part of the same earth system, hence not completely independent. Nonetheless, they can be treated as being independent for the purpose of this study, because of the relatively short time span covered by GRACE observations.

10) *Lines 80 – 85 Section 2.2. As described, this is not an ensemble. No real statistics can be derived from these 4 forward models. One could say: The spread is … and we assert that it is representative of the one sigma about some average. But there is utterly no statistical significance to these. And, by the way, each of the models have their own mountains of data that both support and bias them, and that have data gaps and under/oversampling, none of which is being treated in your empirical procedures.*

We have addressed this issue in point I). We assume that the effect of biases in the original ice histories will be reflected in the uncertainties resulting from the new ensemble.

11) *Line 89, Section 2.2. "think" -> "thick".*

Corrected.

12) *Lines 104-108, Section 2.3. This entire explanation of the procedure to deal with rotationally related surface mass change and GIA has to be redone. I suggest writing out the terms in equation form: What exactly in the end is solved for, and what is the uncertainty? I am completely lost, and if I read the logic – verbatim – I would be led to the conclusion that the paper is simply wrong. I have faith that the author's*

*understanding of all this is more-or-less sound, so I think it is just a series of poor explanation(s) that is involved. It's really a weak part of the paper, however.*

*13) Lines 109-111, Section 2.2. This statement seems contradictory. If not included, why discuss it here and later in the paper? I am quite confused.*

*14) Lines 125-129, Section 3. Again, we return to this confusion about pole-tide and its appearance in the fingerprint maps. What does this mean, or does it even have a meaning? Is it perhaps an artifact? I am not criticizing, I am just confused.*

We have addressed issues 12-14 in point II) and we will amend the text accordingly.

*15) Lines 137-139, Section 3. The sentence starting "The uncertainty overall… " is disturbing, for this is exactly where an empirical model, if at all valuable, might help advancing science. I return to this criticism below, with respect to the 'coupling' to hydrological signal.*

Thanks to the use of a larger ensemble, discussed in point I), the error plots will be considerably different (as shown by Figure R2).

*16) Lines 149-153. Section 3. The proposed explanation for the round region in south Atlantic seems very speculative: a trend in TWS not removed in the pole-tide correction. (?) There should be more explanation to justify this. Why not cumulative errors propagated from GRACE RL06, including the de-aliasing models? Since no error propagation analysis was conducted here, then how do we assess this assertion?*

This explanation was rather concise, hence possibly not clear. We knew that we could not reproduce a large portion of the trend in the C21 and S21 GRACE coefficients, which are directly related to polar motion. This because we had expressly excluded the rotational feedback from the present-day fingerprints. Besides, by "water layer" (line 150), we actually meant to refer to the cryosphere as well, where the Greenland Ice Sheet is by far the largest driver of ongoing polar motion. We did not consider the residual to be possibly related to a mismodelling of the GIA contribution, since the direction of the residual polar motion was not consistent with a GIA source.
However, none of this is relevant anymore, due to the updated treatment of polar motion, as discussed in point II).

*17) Lines 155-156. Section 3. "At the same time, the uncertainties provide…". I suggest the same comment that was mentioned concerning the vagaries of the conclusions. Now those apply here as well. I elaborate on the criticism. I refer to a recent paper by Jensen, L., Eicker, A., Dobslaw, H., Stacke, T., & Humphrey, V. (2019). Longterm wetting and drying trends in land water storage derived from GRACE and CMIP5 models. J.Geophys. Res.: Atmospheres, 124. https://doi.org/10.1029/2018JD029989. In Figure 1 of this paper we see a map of the TWS trends from ITSG-Grace2018s. In that map, note the region of negative (_-10 to -20 mm/yr equivalent water) to the west and south of Hudson Bay, and that also rings across northern Canada in red, and the outer blue ring at amplitude 5 – 15 mm/yr outside of these. Are those really*

*TWS, or are they mismodeled GIA? We can deduce that such questions should apply just by looking at the Jensen 2019 map. So, what, if anything, has the current paper submitted to ESD contributed to this? Has it (the present paper) just affirmed what we surmise from Figure 1 of Jensen?*

We are very thankful for this comment, since we were not aware of that paper.
We have prepared Figure R4, similar to Fig.1 in Jensen et al. (2019), and indeed our results are very different to the west and south-west of Hudson Bay: the large negative spots in Jensen et al. (2019) are actually positive (about 10 mm/yr, at comparable spatial scales) and significant (uncertainty generally smaller than 5 mm/yr in the areas where the signal peaks at more than 10 mm/yr). Notably, in the area west of Hudson Bay, our solution is also more than 10 mm/yr smaller than ICE-6G predictions. In other words, our results suggest that the GRACE-based solution of Jensen et al. (2019) around Hudson Bay is likely biased by mismodelled GIA. This will be discussed in the revised version of our paper.

[Figure]

Figure R4: TWS trend from the new ensemble solution.

*18) Lines 158-160, Section 3. A 300 km buffer is employed. But no Gauss filter is mentioned, so I assume it is not used, and all the signal generated by fingerprint EOFs. What does the buffering do? Some of that buffering will get rid of real signal, not just gravitational artifact, as was the minor point being made by Sterenborg.*

We realise we have not been clear. We note that, for the purpose of obtaining spatially integrated estimates, limits in GRACE resolution often require the use of buffer zones (or other forms of integration kernels), even when no smoothing is applied. Besides, we will revise the text to be more specific about the content of Table 1.
Table 1 listed seven lines. Three of them (Glaciers, Greenland and Antarctica) represent the crysopheric contribution to GMSL: the values listed are obtained directly from the scaling

factors of the fingerprints (which are generated from a known surface load change), in this way avoiding any leakage/buffer issue, since no spatial integration of GRACE fields is actually preformed. The TWS contribution could not be quantified in the same way, since we made use of the EOF fingerprints by Rietbroek et al. (2016). However, considering that the signal from continental hydrology is spread over a very large area, and mostly not in coastal regions, little signal would be lost by the use of a buffer (we have verified that the results are unchanged for buffer sizes of 200-300 km, and only 20% smaller when not using any buffer at all).

The remaining three lines represented: the sum of the previously mentioned contributors (GMSL), and two terms specifically related to GRACE-based GMSL estimates, i.e., the GIA contribution and the residual signal. The GIA contribution was meant to show how much GRACE-based GMSL trend estimates need a GIA model (65% of the total GMSL, in the new ensemble). The residual signal was meant to show which portion of the original GRACE trend remained unexplained: additional tests have revealed that this last number is actually highly depended on the buffer size (with the new ensemble, it reduces to zero when using no buffer at all), so it contains little information and will be removed from the updated table.

> *19) Lines 165-167, Section 3. The estimate of GIA signal is compared to that of Tamisiea. But this is difficult to make much of, as the statement is too equivocal: It is better to state it quantitatively: "We estimate a GIA [thing] of x.x ± y.y, whereas Tamisiea estimated b.b ± c.c, for that same GIA [thing]."*

Agree. We will be more quantitative: our estimate of the GIA contribution to global mean ocean mass change trend, expressed in term of equivalent water height, is 0.8 ± 0.5 mm/yr, whereas Tamisiea (2011) estimated values between 0.8-1.7 mm/yr.

> *20) Lines 175-178, Section 3. A lot is made here of determining a partition of the J2 signal, and convincing (and seemingly rigorous) work was established in Sun et al (2019). But this disambiguation, as reported here, seems notably unconvincing without evaluating error propagation, or at minimum, estimation.*

Agree. As discussed in point I), we have now determined uncertainties from a larger ensemble. We will also more specifically refer to Sun et al. (2019).

> *21) Lines 190-192, Section 4. The statement: "… by using only one dataset we get… on the final solution", is a good one, and maybe in a Brevia paper to this journal that explicit point can be made, even convincing some that it is important! But it is quite challenging to recast this work into something that would convince us that science is being advanced, even by a small epsilon.*

We certainly hope that the Referee will appreciate the revised manuscript. About the suggestion of preparing a Brevia, we think we have already produced a rather concise manuscript. Nonetheless, we believe that all figures and tables are necessary to convey our message, and those would not fit in an even shorter manuscript.

*22) Lines 201-207, Section 5 and final remarks. Again, the big deficit to this paper is that lack of any attention to error propagation, as I suspect that if that were done a similar, but quite useful quantitative conclusion might be discovered. Such quantification could become a valuable thing, especially with respect to planning the next generation of space gravimetry missions.*

We thank the Referee for those encouraging words and again hope that they will be satisfied by the revised manuscript.

Kind regards,
Riccardo Riva and Yu Sun

---

## Author Comment (AC3) · 13 Dec 2019

Response to comments by Referee 3 (Lambert Caron).

We are thankful to Dr. Caron for his positive comments about our study. For a general discussion of the main changes that we have made to the original work, in particular with respect to uncertainty assessment, we refer to the first part of our rebuttal to Reviewer 2. In the following, we address his point-by-point comments (reproduced in italics).

Major points:

1- *l39: I know that other authors have used these terms in previous papers in a somewhat interchangeable manner, but I think it is important to distinguish data-driven from empirical. GIA models derived from partial differential equations (e.g. using love numbers) are not empirical (they are based on a physical theory) but they can be data-driven if their parameters are inverted from a dataset. Among such models are for example Peltier et al. (2015), Lambeck et al. (2014) or Caron et al (2018). Because the authors use such theory to generate their fingerprints, I would argue that their approach is not empirical (and I believe the title should reflect that), and in fact amounts to rescaling the loading history via the least square coefficients as was done in the aforementioned papers, and others before them. In my opinion, that is something the authors could put forward as an advantage of their approach, as it means it is consistent with how we otherwise model and understand the physics behind surface loading and deformation of the Earth interior.*

We have indeed interchangeably used the terms "empirical" and "data-driven" in order to differentiate our approach from previous studies. We also agree that, since our fingerprints are based on an analytical relation between surface load changes and earth mechanical properties, the term empirical is probably too strong.
However, since the fingerprints represent the combined effect of ice mass changes through time and earth rheology (plus a pseudo-spectral solution of the sea level equation), we disagree with the statement that our approach is equivalent to rescaling the loading history as done in some of the cited papers. In other words, even if our results could be used to guide a revision of the input ice histories (see rebuttal to comment #17 by Reviewer 2), we cannot quantify by how much, nor decide whether this revision should also include a change in the assumed mantle viscosity.
Hence, we are going to change the term "empirical" into "semi-empirical", and expand the discussion about the difference between this and other approaches. As suggested, we will emphasize more fact that the fingerprints are based on fundamental physics.

2- *l39 (cont'd) In particular, it means we could compare the GIA scaling coefficients (here the inverted coefficients of the fingerprints) with the values found in the literature and that are based on inverting RSL, and other datasets. That exercise cannot easily be done with true empirical models as they are not built on comparable basis functions. An important question this paper could (begin to) illuminate by showing these coefficients is therefore: are GIA models preferred by GRACE statistically different from the ones constrained with traditional datasets?*

Considering that we somehow disagree on the physical interpretability of our model results, we are not sure how the reviewer meant to realise such a comparison. Nonetheless, we will add a specific comparison against two available global models: Peltier et a. (2015), and Caron et al. (2018). Geoid trend differences are reproduced in Figure R5 and R6 below. We note here that our results are generally smaller than both cited models, though closer to ICE-6G(VM5a), and that the residual patterns are very different between the two cases.

[Figure]

Figure R5: geoid height trend resulting from the difference between ICE-6G(VM5a) and the updated ensemble from this study.

[Figure]

Figure R6: geoid height trend resulting from the difference between Caron et al. (2018) and the updated ensemble from this study.

3- *l81: What is the impact of the number of evaluated cases (here 4) on this statistical analysis? Would the authors expect a lot of differences from a more comprehensive exploration of the parameter space (particularly the viscosity profile)? How much does this limit the applicability of these results to correct GRACE?*

This comment is not applicable anymore, since the new ensemble is based on 132 cases. We would like to note that the new solution is fairly similar (the largest difference being in polar motion, mostly due to the improved approach used for the new solution), but the uncertainties are significantly different.

4- *l88: Ice histories such as that of ANU and ICE-6G_C have been crafted such that when combining all of their regional components, they are able to explain paleo RSL data (especially through the eustatic sea level curve). By recombining regional components of different models, is the solution still consistent with what we know about past RSL - and therefore part of what validates these ice histories in the first place?*

We do not think that we are able to answer this question, nor should we be, and that is why we consider this model to be semi-empirical. A manner to answer this question would be to repeat the inversion that has led to the input ice histories, using our solution as an additional and strong constraint on present-day geoid rates. Such an effort is beyond the scope of this paper.

5- *l113: The authors unfortunately do not really elaborate on their uncertainty quantification approach, and only state that they combine all 4 solutions into an average. How did the author calculate their standard deviation map? Did they: a) take the least-square optimized signal of each of the 4 cases, and then calculated the standard deviation between them (which the first sentence at l119 seems to point to), b) calculated the variance/covariance matrix of the coefficients for each case from the least-square system, which using the notations of Yun et al. (2019) should be a term with a form along the lines of $(F'T'PTF)^{-1}$, and then averaged that covariance matrix between the 4 cases, c) a method similar to b), with a weight associated with each of the 4 cases in the averaging process to take into account that some of them allow smaller residuals than others, d) use yet another method? Out of these possibilities, a) is not an appropriate estimator, it would underestimate the uncertainty as it neglects the level of constraint of each least-square inversion. One could imagine a situation where all 4 best fit produce a similar signal for a given grid point or Stokes coefficient, but with a high variance/low confidence for that value. b) assumes that all 4 cases should have the same weight, which would be acceptable if they yield a similar sum of the residuals, c) being be more indicated otherwise. As this explanation is missing, it is difficult for me to understand and critically examine the results section of the manuscript, and going back to Yun et al. (2019) which details the method, I could not find the information related to uncertainty quantification either. I would add that if the authors mean to provide their model to the GRACE community for correcting GIA, it is very important that the treatment of uncertainty quantification be transparent*

Indeed, we had computed the model uncertainties based on option a). That is still what we do, in the sense that we compute mean and standard deviation of the ensemble. However,

we think that the new uncertainties (shown in the reply to Reviewer 2, Figure R2) are more realistic, considering that the new solution is based on a much larger ensemble, in turn generated by using a rather wide spectrum of viscosity values for the upper and lower mantle.

6- *l179-182: An additional benefit of showing the covariance matrix of the least-square coefficients is that one can verify the degree of independence (or some measure of it, at least) between the different fingerprints by transforming it into a correlation matrix. This way sufficient orthogonality does not have to remain an assumption.*

As also suggested by Referee 1, we have prepared a correlation matrix (see Figure R1 in the rebuttal to Referee 1). We believe that this new figure shows that the fingerprints are quite orthogonal.

Minor points:

7- *l8: if the authors are referring to RSL indicators, they only point to a local level, not global*

True. We meant "globally distributed RSL indicators".

8 - *l35: This reference should be Caron et al. 2018, not 2017*

Corrected.

9- *l78: Why do the authors assume the Earth to be compressible for the fingerprints of the previous section but incompressible for GIA deformation? Is this not inconsistent?*

Elastic fingerprints need to be compressible, otherwise signals in the near field of load changes are unrealistically small. Since present-day GIA fingerprints only reflect mantle relaxation (no near-field load changes, apart from slowly changing ocean loading), the difference between compressible and incompressible solutions on gravity changes is actually quite small (Tanaka et al., 2011, GJI 184). At the same time, incompressible models of viscoelastic relaxation are numerically more stable. Hence, we prefer the possible presence of a small inconsistency against the risk of larger systematic errors in model output. Besides, considering that we now use a larger ensemble, consistency between the two classes of fingerprints (GIA and present-day) is not a major issue: if any is present, it will contribute to the ensemble error.

10- *l81: It was not clear for me at first read whether the authors were combining ICE-6G in one region with another model in the other region, despite the previous sentence. I suggest rewording along the lines of: "we use either GLAC1D (Tarasov et al. 2012) and ANU (Lambeck et al. 2010) in North America and Northern Europe, respectively, or ICE-6G_C (Peltier et al. 2015) in both regions."*

Indeed, that is what we meat. The sentence will be modified as suggested.

11- *l81: "of" should read "or"*

Corrected.

12- *l88: The authors reference Ivins & James (2005) for the IJ05 model, which had an updated version (IJ05_R2) released in 2013 (Ivins, E. R., T. S. James, J. Wahr, O. Schrama, J. Ernst, F. W. Landerer, and K. M. Simon (2013), Antarctic contribution to sea level rise observed by GRACE with improved GIA correction, Journal of Geophysical Research: Solid Earth, 118(6), 3126–3141). If the authors used the updated version, this is simply a matter of updating the reference, but if not I would be curious to know why they chose the old version and if they expect a significant change from this choice. The volume of the Antarctic ice sheet at the LGM is different by about a factor 2 for example.*

We were actually not aware of such a large difference in ice volume at LGM between IJ05 and IJ05_R2. Nonetheless, our GRACE-only approach cannot properly solve for Antarctic GIA, due to the large spatial extent and rather linear temporal evolution of ice sheet mass change, as well as its spatial overlap with GIA signals.
Hence, we have decided to stick to our initial approach, which is to use a single Antarctic fingerprint that had been roughly calibrated against the GRACE-ICESat combination by Riva et al. (2009). In that sense, it does not matter what the ice history actually is, since it is the GIA geoid signature that is used here (i.e., the combined effect of ice evolution and viscosity structure). We note that we still allow this fingerprint to be scaled within the inversion, but that only affects the actual magnitude of the signal, not its spatial pattern.

13- *l116: "rounder": do you mean smoother or with a more circular shape?*

We meant of a more circular shape, but this has actually changed in the new ensemble solution (the largest differences with respect to ICE-6G in North America are now over and West of Hudson Bay, see Figure R5), so the whole sentence will be modified.

Kind regards,
Riccardo Riva and Yu Sun